# HIV prevalence, risk factors, prevention methods, and interventions among taxi drivers and commercial motorcyclists in sub-Saharan Africa: A scoping review

John Baptist Asiimwe[1]*, Benjamin Betunga[2], Lilian Birungi[3], Joy Kabasindi Kamanyire[3], Moses Wankiiri[1], Lilian Nuwabaine[4], Joseph Kawuki[5], Edward Kumakech[6]

1 School of Nursing and Midwifery, Aga Khan University, Kampala, Uganda, 2 The AIDS Support Organization, Tororo, Uganda, 3 Sultan Qaboos University, Oman, 4 School of Midwifery, Makeni, and Seed Global Health, Sierra Leone, 5 Program in Public Health, Department of Family, Population, & Preventive Medicine, Stony Brook University, Stony Brook, New York, United States of America, 6 Faculty of Nursing and Midwifery Lira University, Lira, Uganda

* john.asiimwe@aku.edu

## Abstract

Transport workers, particularly taxi drivers and commercial motorcyclists, comprise a population at high risk for HIV and account for nearly one-third of new HIV infections in sub-Saharan Africa (SSA). Transport workers bridge HIV infections from high-risk populations to the general population. This scoping review aimed to map the available evidence around HIV prevalence, risk factors, prevention methods, and interventions among taxi drivers and commercial motorcyclists in SSA. This scoping review used the Arksey and O'Malley framework. Published articles were retrieved from MEDLINE, CINAHL, African Index Medicus, Web of Science, Scopus, EMBASE, HINARI, and Google Scholar from January 2000 to August 2024. Two authors screened the titles and abstracts of retrieved studies and examined the references of relevant articles for additional literature. Three authors independently extracted data from the included studies using a standard data extraction form. Data were analyzed using descriptive statistics and content analysis techniques. This review included 24 out of 126 studies. The HIV prevalence was 2.02%–9.9% among commercial motorcyclists and reached 33.4% in samples comprising both motorcyclists and taxi drivers. The high HIV infection rate in SSA when compared with the global adult general population was associated with numerous behavioral (e.g., multiple sex partners), psychosocial (e.g., stigma), and sociodemographic (e.g., age) risk factors. However, there was suboptimal use of HIV prevention methods such as safe male circumcision (20.7%–64.9%) and condoms (26%–45.7%) and few interventional studies (n = 2). Despite HIV testing being an entry point for chronic care, we found no study reporting the HIV cascade for commercial motorcyclists or taxi drivers with HIV. To inform

**Data availability statement:** All relevant data are within the paper and its Supporting Information files.

**Funding:** The authors received no specific funding for this work.

**Competing interests:** The authors declare that no competing interests exist.

better HIV policies and programs in SSA this review recommends additional observational and interventional research on HIV incidence, predictors, new models of HIV testing, antiretroviral-based HIV prevention methods, and the role of peer-to-peer support models in reducing HIV infection.

## Introduction

Despite a 39% reduction in new HIV infections since 2010, with infections dropping from 2.1 million to 1.3 million in 2023, this figure still falls significantly short of the global goal of fewer than 370,000 new infections by 2025 [1–2]. Similarly, while the rate of HIV infection in sub-Saharan Africa (SSA) is decreasing, HIV continues to be a significant healthcare challenge. Two-thirds of people living with HIV worldwide (25.5 of 38.4 million) are from the African region, which increases the risk for re-infection and cross-infection [1]. Despite some progress in reducing the HIV incidence in SSA [2], key populations at a high risk for HIV and their partners (e.g., sex workers, men who have sex with men [MSM], and people who inject drugs) continue to represent the largest proportion of new HIV infections. For example, key populations and their partners accounted for 55% of the 1.3 million new infections in 2023, which exceeded the 51% contribution in 2021 [2,3]. Transport workers, particularly taxi drivers and commercial motorcyclists, make up 70% of clients of the key population (e.g., sex workers), and account for 26% of new HIV infections in SSA [1,3,4].

Taxi drivers and commercial motorcyclists form the majority of transport workers in SSA and their economic contribution is undisputed. These transport businesses provide a source of employment for youth and provide an income or livelihood for many families in SSA [5]. However, research indicates that taxi drivers and commercial motorcyclists are major contributors to HIV transmission in SSA [5,6]. The mobile nature of transport work, availability of cash funds, and ability to pay for sex expose them to a wide network of sexual partners whereby they can potentially contract and spread HIV [7,8]. Taxi drivers and commercial motorcyclists were reported to engage in unprotected sexual intercourse with both low- and high-risk groups, including sex workers, schoolgirls, market women, female hawkers, and married women, some of whom may be their customers [7]. Therefore, taxi drivers and commercial motorcyclists have been described as a "bridging population," who bridge HIV infections from key populations at high risk to the general population [5,7].

A clear understanding of the HIV epidemic is necessary to design multifaceted strategies to reduce HIV among taxi drivers and commercial motorcyclists and achieve the 2030 global target of a 90% reduction in new HIV infections [2]. This includes clarifying the burden of HIV, HIV risk factors, use of HIV prevention services, and interventions that have been successful in reducing the HIV burden in this group. However, the HIV epidemic among taxi drivers and commercial motorcyclists remains unclear. Previous reviews explored the burden of HIV and HIV programs among long-distance truck drivers, but did not report the HIV burden among local transport workers in SSA [9–12]. Therefore, this study sought to map the available evidence around the HIV prevalence (i.e., percentage of those diagnosed with HIV),

risk factors for HIV infection, use of HIV prevention methods (e.g., safe male circumcision [SMC]), and HIV interventions among taxi drivers and commercial motorcyclists in SSA. Identifying gaps in the literature will guide further research and efforts (interventions) to reduce the burden of HIV among transport workers in SSA.

## Methods

A protocol for this study was developed and registered with an open-source foundation (https://doi.org/10.17605/osf.io/z7dyv). The review was conducted and reported following the Arksey and O'Malley framework [13], which provides five clear and comprehensive steps for conducting scoping reviews. These steps are: 1) identifying the research question; 2) identifying relevant studies; 3) study selection; 4) charting the data; and 5) collating, summarizing, and reporting the results. Consistent with the established methods for conducting scoping reviews, this study did not assess the method-ological quality (critical appraisal) of the included studies.

### Identification of the research questions

This review focused on addressing the following four research questions related to the HIV epidemic among taxi drivers and commercial motorcyclists. 1) What is the HIV prevalence among taxi drivers and commercial motorcyclists in SSA? 2) What are the risk factors for HIV infection among taxi drivers and commercial motorcyclists in SSA? 3) What proportion of taxi drivers and commercial motorcyclists in SSA use specific HIV prevention services (e.g., condoms, antiretroviral ther-apies, pre-exposure prophylaxis [PREP], and post-exposure prophylaxis)? 4) What interventions have been implemented to increase the use of HIV prevention services or reduce HIV among commercial motorcyclists and taxi drivers in SSA?

### Identification of relevant studies

We searched titles, abstracts, and MeSH terms in several databases including MEDLINE (PubMed), SCI-Finder, CINAHL, African Index Medicus, Web of Science, Scopus, EMBASE, HINARI, and Google Scholar, from January 2000 to August 2024. We also searched for gray literature in ProQuest Dissertations and Theses, Global WorldCat Dissertations and Theses, and the Google search engine. After the initial search, we examined the references of relevant articles to locate additional literature. The search strategy was developed by the authors with the help of a librarian and pilot-tested with five articles before the main search (Table 1, S1 Table).

Table 1. Search strategy.

| # | Search terms | |
|---|---|---|
| 1 | Taxi drivers OR motor vehicle taxi drivers OR minibus taxi drivers OR minibus drivers OR Boda Boda riders OR Boda Boda operators OR Boda Boda motorcycle drivers OR Boda Boda motorcyclists OR motorcycle drivers OR commercial motorcycle drivers OR tricycle drivers | AND |
| 2 | HIV OR acquired immune syndrome OR HIV-2 OR HIV-1 OR HIV infections | AND |
| 3 | Prevalence OR frequency OR extent OR rate OR proportion OR percent* OR seroprevalence OR HIV serodiagnosis OR HIV seropositive* OR HIV seronegative* or HIV test* | AND |
| 4 | Predictors OR associated factors OR correlates OR risk factors OR risky sexual behaviors OR unsafe sex OR risk taking OR health risk behaviors OR sexual behaviors | AND |
| 5 | Condoms OR condom use OR condom programming OR HIV testing OR AIDS testing OR voluntary counsel* OR self-testing OR diagnosis OR point-of-care testing OR mass screening OR HIV testing and counsel* OR serologic test OR pre-exposure prophylaxis OR PREP OR antiretroviral agents OR antiretroviral therapy OR HIV infections/drug therapy OR highly active OR anti-HIV agents OR circumcision OR safe male circumcision OR male circumcision OR SMC OR use OR utilize* | AND |
| 6 | Intervention OR strategy* OR plans OR strategic plans OR improve* OR advances OR policy* OR approach* | AND |
| 7 | Limited to English language and papers published between 2000 and 2024 | AND |
| 8 | **#1 AND #2 AND #3 AND #4 AND #5 AND #6 OR #1 AND #2 AND #3 AND #6 OR #1 AND #2 AND #5 AND #6** | |

This review included studies conducted among taxi drivers and commercial motorcyclists in SSA that reported HIV prevalence, HIV risk factors, use of HIV prevention services, or HIV interventions. Taxi drivers and commercial motorcyclists in these studies may or may not have self-reported their HIV serostatus or use of HIV prevention services. Only studies published in the English language were included. We included all population-based studies conducted in community-based settings, and both qualitative and quantitative studies were eligible. Studies reporting on truck drivers, along with editorials, book chapters, expert opinions, conference abstracts, and systematic reviews were excluded from this review. In addition, we excluded studies that did not disaggregate data across the study groups, such as those that included truck drivers in the analysis of HIV prevalence [4].

### Study selection

The retrieved articles were screened against the inclusion criteria. After searching the databases, we imported all retrieved articles into EndNote and grouped them by the database of origin. Duplicated articles were then removed. Next, two reviewers (BB, AJB) undertook title and abstract screening for all retrieved articles. After screening, the full-texts of the selected articles were retrieved and two reviewers independently confirmed if they met the inclusion criteria. Articles were read in detail three times. A PRISMA flow chart was used to document the article selection process (Fig 1). Any disagreements between the two reviewers at this stage were resolved by a third reviewer (JK).

### Data charting/extraction

Three reviewers (BB, AJB, BL) independently extracted the data using a standard data extraction form (S2 Table). The data extraction form was created by the authors based on Arksey and O'Malley [13], and collected study characteristics (e.g., study year), methodology (e.g., design), main concept or themes reported, key findings as per the research questions, study limitations (per study), study gaps, conclusions, and future directions or recommendations. The data extractors were trained on extracting data from the retrieved articles. The data extraction form was independently pilot-tested with five articles of various types. Data were extracted twice by the same authors and consensus was reached through discussion.

### Collating, summarizing, and reporting the results

Data for the study characteristics were entered, coded, and analyzed in SPSS (version 20) using descriptive statistics such as frequencies, and then presented in figures. Qualitative data were analyzed using content analysis techniques [14]. We extracted the study findings and their corresponding quotations and inserted them into a Microsoft template for comprehensive reading. Specifically extracted data were deductively organized around the study objective or theme (e.g., risk factors for HIV infection) in Microsoft Word 2010. Patterns were then identified in the text and color-coded to form categories and codes. Data were presented in narrative form around four themes: HIV prevalence, HIV risk factors, use of HIV prevention services, and HIV interventions. Each theme was presented alongside key subthemes and categories. The Preferred Reporting Items for Systematic reviews and Meta-Analyses extension for Scoping Reviews (PRISMA-ScR) Checklist was used to report the findings (S1 PRISMA-ScR Checklist).

## Results

### Study flow

In total, we retrieved 151 articles (Fig 1). After eliminating duplicates, 126 articles underwent title/abstract screening, and relevant studies were identified for full-text screening. Finally, 24 studies that met the inclusion criteria were included in this review.

## Characteristics of the included studies

The included studies comprised 21 research papers and three thesis reports, and were published between 2005 and 2023. Nearly half (n = 11) were published between 2019 and 2024 (Table 2). Most studies were conducted in the East African region (Fig 2). Across countries, most studies were conducted in Uganda (n = 8), followed by Nigeria (n = 6) and South Africa (n = 5). The majority of the studies were quantitative and used cross-sectional (n = 18) and interventional designs (n = 2), three were qualitative (cross-sectional designs), and two were mixed methods' studies. All studies were conducted in the community, among men, and most involved commercial motorcyclists (n = 16). Many studies reported on the use

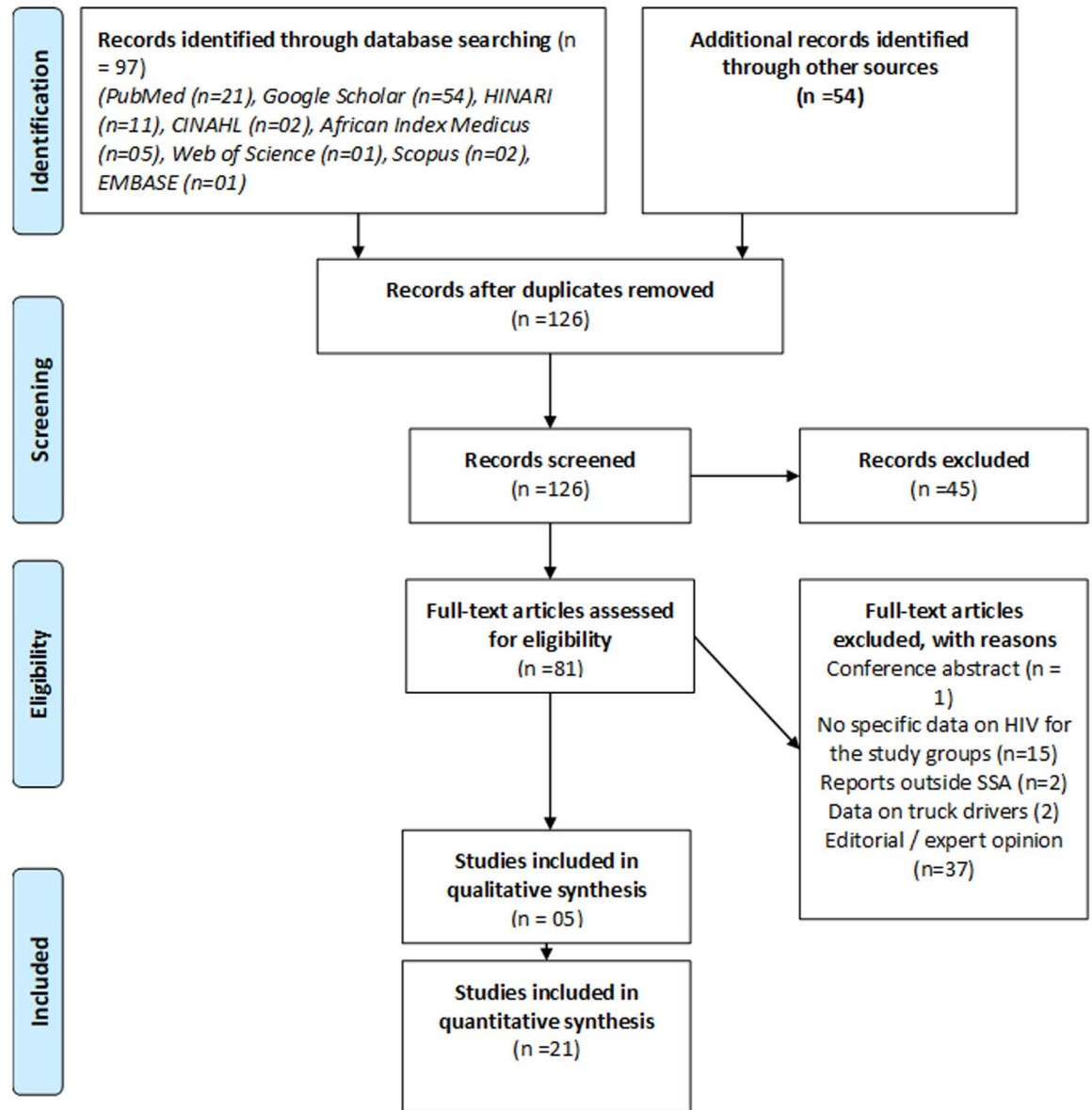

**Fig 1. Study selection process.** Two studies provided qualitative and quantitative findings (mixed methods); thus, these papers have been captured under quantitative and qualitative synthesis.

**Table 2. Characteristics of the studies included in this scoping review.**

| Author | Publica-tion year | Study year | Coun-try | Study design | Thematic/sub-thematic areas | Sample size | Participants' age (years) |
|---|---|---|---|---|---|---|---|
| Lakew & Tamene | 2014 | 2006 | Ethiopia | Cross-sectional, quantitative | HIV risk behaviors and associated factors; use of HIV prevention services | 615 | 15–45 |
| Challe et al. | 2022 | 2016 | Tanza-nia | Cross-sectional, quantitative | HIV prevalence and risk factors, HIV knowledge, atti-tudes, and practices; use of HIV prevention services | 973 | 18–59 |
| Lindan et al. | 2015 | 2004–2005 | Uganda | Cross-sectional, quantitative | HIV prevalence and risk factors; use of HIV preven-tion services | 683 | 15–49 |
| Ssekankya et al. | 2021 | 2020 | Uganda | Cross-sectional, quantitative | Use of HIV prevention services; HIV knowledge, risk; stigma | 305 | ≥18 |
| Tumwe-baze et al. | 2020 | 2019 | Uganda | Cross-sectional, quantitative | HIV prevalence and risk factors; HIV knowledge; use of HIV prevention services | 375 | 17–42 |
| Tusabe et al. | 2022 | 2020 | Uganda | Cross-sectional, mixed methods. | Use of HIV prevention services and related beliefs | 316 | 18–49 |
| Mchunu et al. | 2020 | 2013–2015 | South Africa | Cross-sectional, qualitative | Use of HIV prevention services | 2 FGD (18 participants) | Not specified |
| Mchunu et al. | 2012 | Not specified | South Africa | Cross-sectional, quantitative | HIV knowledge/perceptions; use of HIV prevention services; HIV risk behaviors | 2 FGD (8–10 participants) | Not specified |
| Moham-med et al. | 2019 | 2018 | Nigeria | Cross-sectional, quantitative | HIV prevalence | 379 | 16–50 |
| Adelekan et al. | 2017 | 2016–2017 | Nigeria | Interven-tional study, quantitative | HIV prevalence; use of HIV prevention services | 2878 | Not specified |
| Ojieabu & Eze | 2011 | Not specified | Nigeria | Cross-sectional, quantitative | HIV knowledge; HIV risk behaviors; use of HIV prevention services | 345 | 20–59 |
| Muhamed et al. | 2007 | Not specified | Nigeria | Cross-sectional, quantitative | HIV prevalence | 379 | 16–50 |
| Ncama et al. | 2013 | 2009 | South Africa | Cross-sectional, Quantitative | HIV knowledge, beliefs, and practices/behaviors; use of HIV prevention services | 175 | 18–59 |
| Olarewaju et al. | 2012 | 2007 | Nigeria | Cross-sectional, quantitative | HIV knowledge, beliefs, and behaviors; use of HIV prevention services | 298 | ≥15 |
| Mandela et al. | 2022 | Not specified | Uganda | Cross-sectional, quantitative | HIV knowledge, attitudes, and behaviors; use of HIV prevention services | 296 | ≥18 |
| Akinyi & Otengah | 2018 | 2017 | Kenya | Cross-sectional, quantitative | HIV prevalence; use of HIV prevention services | 122 | 20–40 |
| Betunga et al. | 2023 | 2022 | Uganda | Cross-sectional, quantitative | HIV risk behaviors | 420 | 18–55 |
| Nyanzi et al. | 2005 | 2000–2001 | Uganda | Cross-sectional, qualitative | HIV-related beliefs; HIV risk | 212 | 17–40 |
| Potgieter et al. | 2012 | Not specified | South Africa | Cross-sectional, mixed methods | HIV risk behaviors | 223 | ≥26 |
| SiuLapwa | 2006 | Not specified | Zambia | Cross-sectional, quantitative | HIV prevalence, HIV risk factors, and use of HIV prevention services | 359 | ≥20 |
| Mlughu et al. | 2020 | Not specified | Tanza-nia | Cross-sectional, qualitative | Use of HIV prevention services | 35 | 18–24 |
| Mokgatle | 2020 | 2016 | South Africa | Cross-sectional, quantitative | HIV knowledge/awareness; HIV risk factors; use of HIV prevention services | 722 | 19–68 |
| Nabifo et al. | 2021 | Not specified | Uganda | Cross-sectional, quantitative | HIV risk factors; use of HIV prevention services | 401 | 18–59 |
| Olarewaju et al. | 2015 | 2007–2008 | Nigeria | Interven-tional study, quantitative | HIV knowledge, attitude and risky sexual behavior | 150 | 17–57 |

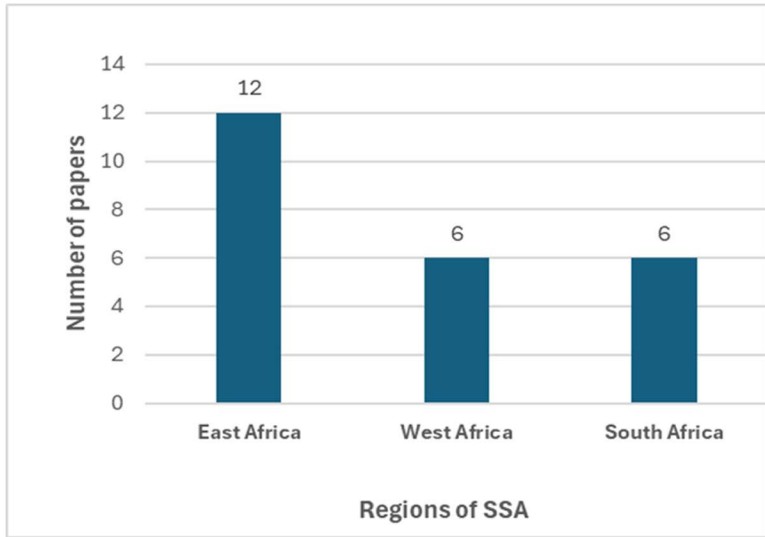

**Fig 2. Number of studies retrieved per SSA region.**

of HIV prevention methods (n = 19) and HIV risk factors (n = 14), with eight studies reporting prevalence and two reporting interventions (Table 2). The sample size in the quantitative studies ranged from 122 to 2878 participants, and that in the qualitative studies from 16 to 223 participants. The majority of the studies included participants older than 18 years (n = 13).

### HIV prevalence

Eight studies reported HIV prevalence estimates for taxi drivers and commercial motorcyclists [5,7,11,15–19]. Overall, the HIV prevalence varied from 2.02% to 9.9% in studies that only considered commercial motorcyclists and reached 33.4% in those that included both taxi drivers and commercial motorcyclists in the analysis. The highest prevalence of HIV among commercial motorcyclists was reported in three studies, all conducted in Uganda (7.43%–9.9%) [5,7,19]. The prevalence estimates in most studies were based on self-reported questionnaire data, except for three studies that reported the HIV prevalence based on standard HIV screening and diagnostic antibody tests [5,15,16] (Fig 3).

Four studies reported that the HIV prevalence among commercial motorcyclists varied by residence/region or area of work, tribe, number of sexual partners, marital status, educational level, alcohol consumption, knowledge of HIV and screening, having a sexually transmitted infection, health-seeking behavior, and age [5,7,15,18]. For example, Lindan et al. [7] indicated that the odds of commercial motorcyclists being diagnosed with HIV were 2.42- and 5.69-times higher among those with genital ulcer disease and those aged ≥31 years, respectively, compared with those without genital ulcer disease or aged 18–20 years. Tumwebaze et al. [5] found that commercial motorcyclists who reported multiple sexual partners had a six-times higher chance of acquiring HIV/AIDS compared with those engaged in buying sex from sex workers. However, the risk of acquiring HIV was lower among married participants and those with regular partners compared with those who were single or without regular partners, respectively [15].

It is worth noting that studies aggregated specific HIV prevalence data for taxi drivers with other risk groups were outside this study scope; these studies are not reported here (4). In addition, no study reported HIV incidence estimates for taxi drivers and commercial motorcyclists.

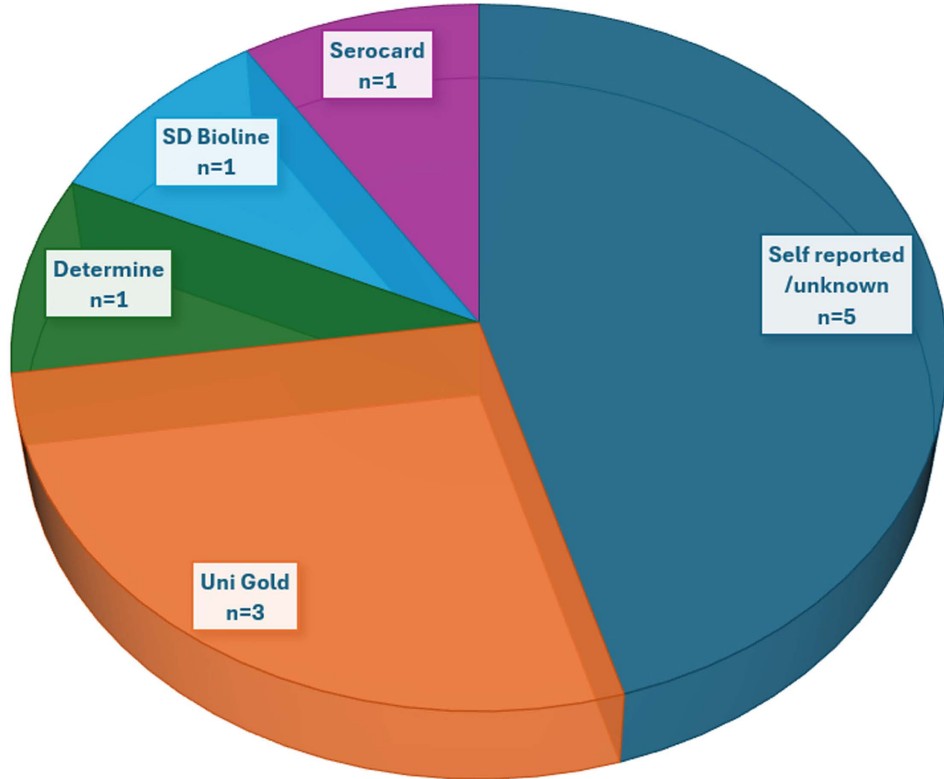

**Fig 3. Methods used to determine HIV prevalence among taxi drivers and commercial motorcyclists in SSA.** It is important to note that the three studies reported the use of more than one HIV determination method. Uni Gold was reported by all three studies. Whereas Determine, SD Bioline, and Serocard were reported by each of the three studies (in addition to Unigold).

## HIV risk factors

HIV risk factors among taxi drivers and commercial motorcyclists were categorized as behavioral (e.g., multiple sexual partners/transactional sex, substance/drug use, condomless sexual intercourse), psychosocial (e.g., HIV knowledge/ awareness/perceptions, perceived HIV risk, stigma), and sociodemographic risk factors (Fig 4). The most commonly reported HIV risk factors were behavioral, followed by psychosocial and sociodemographic risk factors. Notably, few studies investigated sociodemographic risk factors, and no study explored the structural determinants of HIV infection among taxi drivers and commercial motorcyclists in SSA.

## Behavioral risk factors

   **Multiple sexual partners and transactional sex.** The number of sexual partners was evaluated in various ways including having two or more sexual partners (irrespective of gender or sexual orientation), length of contact with partners (regular, permanent, or casual), number of lifetime partners, or number of partners with whom they paid for sex (transactional sex) [7,15]. Overall, the reviewed studies indicated that a large proportion of taxi drivers and commercial motorcyclists were involved in extramarital sexual affairs and had multiple partners, some of whom were their business clients [5,7]. Six studies indicated that 29.3%–74.3% of taxi drivers and commercial motorcyclists had more than one sexual partner, with four of these studies reporting more than half of the participants had extramarital sex within 1 year before the study [4,5,20–24]. A study conducted among Ethiopian

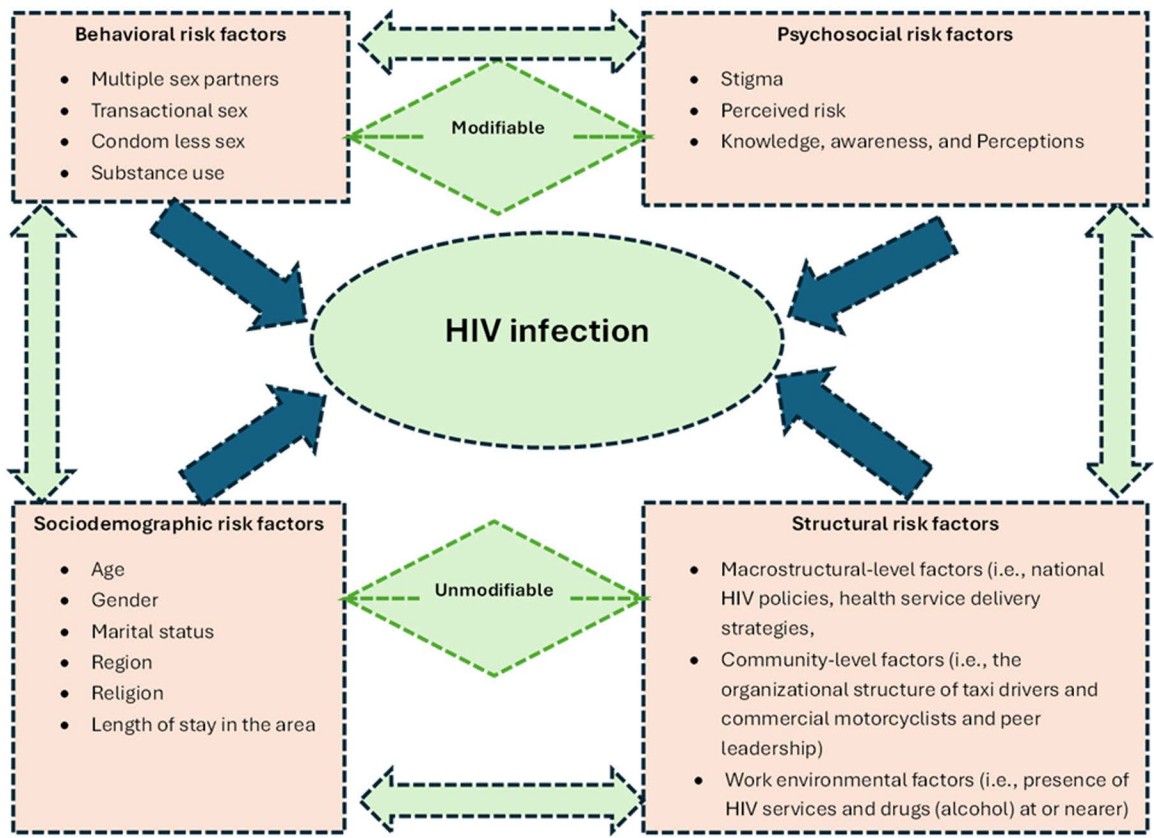

**Fig 4. Conceptual framework indicating multifaceted factors associated with HIV infection among taxi drivers and commercial motorcyclists in SSA.** This indicated that modifiable and non-modifiable factors interacted and influenced HIV infection among transport workers in SSA [5–11,15–31].

taxi drivers reported the average number of lifetime sexual partners was 7.1 and that of partners in the 12 months preceding the survey was 1.6 [8].

Studies conducted in Ethiopia, Tanzania, and Uganda categorized multiple sexual partners as regular, casual, and commercial sexual partners [7,8,15,25]. Overall, regular partners comprised the majority of multiple sexual partners, followed by casual and commercial partners. The proportion of commercial motorcyclists with regular sexual partners was the highest (29%–50.9%) [7,15]. Three studies reported the proportion of taxi drivers and commercial motorcyclists with casual sexual partners (22.8%–34%) [8,15,25]. Lindan et al. [7] reported that 8.7% of commercial motorcyclists had sex with a casual male partner. However, studies conducted in Tanzania, Uganda, and Nigeria found that a relatively lower proportion of commercial motorcyclists reported sexual relationships with commercial sexual partners (7.7%–36%) [7,15,22,25].

Qualitative studies from South Africa and Uganda highlighted reasons why taxi drivers and commercial motorcyclists often engaged in multiple sexual relationships with married or unmarried partners [21,23,26]. These included a belief that married women were in a safe relationship; these women were believed not to have HIV and transport workers could either subsidize or pay the transport fares for married women in exchange for sex [26]. Similarly, there was also a preference for virgin girls/women because of beliefs that they were HIV-free, provided prestige to men in relationships with them, and offered potential for long-lasting relationships [7]. Other reasons were related to deeply entrenched cultural beliefs such as religion (e.g., Islam) that supports polygamy, a history of being raised in a polygamous family, and multiple sexual partners being considered a measure of masculinity and prestige among peers [23,26].

**Substance use.** Four studies assessed substance use as a behavioral risk factor for HIV among taxi drivers and commercial motorcyclists. These studies considered the use of/intoxication with alcohol, khat, or cigarettes before or during the last sexual intercourse or a history of alcohol consumption on a daily, weekly, or monthly basis [7,8,15,27]. Only one study used the Alcohol Use Disorders Identification Test (AUDIT) tool to assess alcohol consumption, drinking behaviors, and alcohol-associated problems [25]. Two studies from Uganda and Nigeria indicated alcohol was the most consumed substance (used by 43%–55.4% of motorcyclists), followed by cigarette smoking (5.9%–21.5%) [22,27]. One study reported the use of khat among taxi drivers in Ethiopia but did not provide the proportion of users [8].

The use of stimulants such as alcohol and khat was thought to motivate taxi drivers or commercial motorcyclists to have unsafe sex and predispose them to HIV. Three studies indicated that drinking alcohol during the last sexual intercourse was associated with HIV [8,15]. A high prevalence of HIV among commercial motorcyclists in Tanzania (3.9%) was found among those consuming alcohol, and they had a 4.5 times odds of acquiring HIV [15]. Nabifo et al. [25] reported that 35% of commercial motorcyclists in Uganda were engaged in sexual intercourse under the influence of alcohol. Notably, a qualitative study found that taxi drivers often drank alcohol when having sexual intercourse with girls [28]. Moreover, the use of stimulants may predispose taxi drivers and commercial motorcyclists to other risky sexual behaviors. For example, harmful alcohol consumption among motorcyclists was found to increase the odds of engaging in other HIV transmission risk behaviors by three-fold, such as having sex with sex workers or condomless sex [25]. Among taxi drivers, not using khat was associated with higher odds of lifetime abstinence and being faithful to their regular partners [8]. Notably, no study estimated the magnitude of the risk of HIV associated with cigarette smoking, chewing khat, or using other drugs (e.g., chewable tobacco or cocaine) among taxi drivers and commercial motorcyclists.

**Condom use during sex.** Five studies conducted among taxi drivers and commercial motorcyclists in Uganda, Zambia, and South Africa indicated that condom use during sexual intercourse was suboptimal (26%–45.7%) [5,18,19,21,24]. In terms of the consistency of use, three studies conducted in Nigeria and South Africa reported that the largest proportion of commercial motorcyclists and taxi drivers used condoms "sometimes" (42.5%–64.6%), followed by "always" (22.4%–39.7%) and "never" (6.7%–35.1%) [20,22,24]. Most taxi drivers and commercial motorcyclists used condoms when they had sex with sex workers (37.4%–86.4%), followed by regular partner (37.4%) and casual partners (7%–22.8%) [8,15]. However, a study from Uganda found that a larger proportion of commercial motorcyclists used condoms during sexual intercourse with casual partners (31.9%) than with regular partners (17.9%) [7]. This was confirmed by a qualitative study [23]. Although knowledge about the role of condoms in the prevention of HIV among taxi drivers and commercial motorcyclists was above average (58.3%–77%), few used them [20,21,29].

Most reasons for the suboptimal use of condoms among taxi drivers and commercial motorcyclists during sexual intercourse were related to individual rather than health system factors. These reasons included myths and misconceptions around condom use (e.g., condoms are meant to depopulate Africa, are porous, and can cause impotence), trust in the partner, partner's dislike of condoms, inadequate or lack of condoms in workspaces, and lack of knowledge about the correct use of condoms [18,19,23,26,29]. One study conducted in Ethiopia found that 58.5% of taxi drivers had no condom use efficacy, with 44% experiencing condom slippage or breakage during sex with casual partners [8]. Furthermore, two studies found that income, age, and use of voluntary counseling and testing (VCT) were associated with condom use among taxi drivers and commercial motorcyclists. Commercial motorcyclists aged ≥35 years (compared with those aged 18–25 years) and taxi drivers who had not used VCT (compared with those who had used VCT) were less likely to use condoms [8,19]. Conversely, taxi drivers with a better monthly income were more likely to experience condom use efficacy than those with low income [8]. Finally, although an association between the risk of HIV acquisition and not using condoms during the last sexual intercourse was reported among taxi drivers, the magnitude of that risk was not quantified [18].

## Psychosocial risk factors

**HIV knowledge, awareness, and perceptions.** Two thematic areas were assessed under knowledge, awareness, and perceptions: HIV transmission and prevention [15,19,20,22,26,27]. One study used an unstandardized 5-item Likert

scale to measure knowledge about HIV transmission and prevention among commercial motorcyclists, with knowledge categorized as either good or poor; 90.8% had good knowledge about HIV [27]. The other quantitative studies used binary responses (yes/no) to HIV-related questions to measure HIV knowledge or awareness [15,19,20,22]. Generally, 97%–100% of commercial motorcyclists were aware of the existence of HIV as a disease [20,22]. Most (89.3%–96.2%) commercial motorcyclists and taxi drivers knew that HIV is transmitted through penetrative sex [15,19,20,22]. A study from Nigeria indicated that commercial motorcyclists knew that HIV is transmitted via sharing injection needles (73.4%), penetrative anal sex (61.2%), blood transfusion (60.3%), sharing shavers (57.1%), and maternal transmission (48.7%) [20]. Similarly, a qualitative study found that taxi drivers knew that HIV is incurable and transmitted via blood and sexual intercourse [26]. However, a study from Uganda found that 62.5% of commercial motorcyclists were not knowledgeable about how to prevent HIV [19].

**HIV perceived risk.** Three studies determined the perceived risk of acquiring HIV among commercial motorcyclists using binary responses (yes/no) to HIV risk-related questions [15,19,27]. In these studies, the majority of commercial motorcyclists (63.85%–89.7%) knew that they were at high risk for acquiring HIV [15,19,27]. In addition, a large proportion of commercial motorcyclists (>71.4%) knew that some of their sexual partners or business clients (e.g., sex workers, bar staff, MSM, and drug abusers) were also at high risk for acquiring or having HIV infection [15]. One study among motorcyclists found an association between perceived HIV risk and having a positive HIV diagnosis, but the magnitude of that risk was not quantified [5]. However, no study measured the perceived risk of acquiring HIV among taxi drivers and its association with HIV infection.

**HIV stigma.** The two studies that evaluated HIV stigma among commercial motorcyclists in Uganda reported contradictory findings, which may be related to how stigma was determined [25,27]. Nabifo et al. [25] used the standardized STRIVE scale that measures HIV stigma and discrimination and found that 82% of motorcyclists had HIV-related stigma. Conversely, Ssekankya and colleagues [27] used an unstandardized 4-item Likert scale to measure HIV stigma, which showed 4.6% of commercial motorcyclists had HIV-related stigma. HIV stigma among commercial motorcyclists was associated with other risky sexual behaviors for HIV infection, such as having sex with sex workers or condomless sex [25]. Commercial motorcyclists with any HIV-related stigma had higher odds of having HIV transmission risk behaviors than those with no HIV stigma [25]. However, there were no qualitative or quantitative studies that explored the facilitators, barriers, or factors associated with HIV stigma or the magnitude of association between HIV stigma among motorcyclists and taxi drivers and HIV infection.

## Sociodemographic risk factors

Three studies conducted among taxi drivers and commercial motorcyclists found statistically significant associations between HIV infection and region, age, length of stay in an area, marital status, and religion [7,15,18]. However, the key predictors of HIV infection were age and marital status. Challe et al. [15] found that married commercial motorcyclists had 20% lower odds of HIV infection than their unmarried counterparts. Motorcyclists aged ≥31 years had 5.7-times higher odds of being diagnosed with HIV compared with those aged 18–20 years [7]. Overall, few studies investigated sociodemographic risk factors, and no study explored the structural determinants of HIV infection among taxi drivers and commercial motorcyclists in SSA.

## Use of HIV prevention services: HIV counseling and testing

The use of HIV counseling and testing services was evaluated by ever having used VCT services in the past or having used these services within a specific period (e.g., the past 3–12 months). The proportion of commercial motorcyclists and taxi drivers who had ever used HIV testing services ranged from 36.9% to 96%, with five of six studies reporting a proportion above 65% [5,7,15,19,24,27]. A large proportion of motorcyclists had received HIV testing services from a health facility (87%) and been tested to know their HIV status (67.6%–85%) [19,27].

Barriers to the use of HIV testing services among commercial motorcyclists were also reported. Mandela et al. [19] found key barriers to HIV testing were HIV stigma, unavailability of HIV testing services, and lack of knowledge about HIV testing. Other challenges to HIV testing among young motorcyclists included a lack of privacy during counseling, fear of HIV status disclosure to others, and a shortage of HIV counselors [30]. One study reported several factors associated with HIV testing among commercial motorcyclists, including age, HIV-related stigma, knowing the HIV status of the primary partner, and good knowledge about HIV [27]. Having HIV-related stigma and being older (≥30 years) were associated with reduced odds of using HIV testing services. Conversely, knowing one's HIV status and having good knowledge of HIV were associated with higher odds of using HIV testing services.

Despite HIV testing being an entry point for chronic care, we found no study reporting the HIV cascade for commercial motorcyclists and taxi drivers who had tested for HIV, including the use of antiretroviral therapy, PREP, and post-exposure prophylaxis. Although the use of a combination of HIV prevention services was found to reduce HIV, only one study was conducted on the use of a combination of HIV prevention services among taxi drivers, truck drivers, and commercial motorcyclists in Uganda [4].

## Use of HIV prevention services: SMC

Three studies reported the uptake of SMC among taxi drivers and commercial motorcyclists, with the proportion of those circumcised ranging from 20.7% to 64.9% [6,18,19]. Reasons commercial motorcyclists used SMC included HIV prevention, religion, and personal hygiene [19]. Similarly, a qualitative study conducted in Uganda among commercial motorcyclists reported the facilitators of SMC were improved penile hygiene, perceived sexual function, reduction in chances of contracting sexually transmitted diseases (e.g., HIV), and health education about SMC [6]. The barriers to SMC uptake included the fear of pain after SMC, fear of knowing one's HIV status because of compulsory HIV testing before SMC, long healing time leading to a delayed return to work, financial loss during the healing time, fear of loss of male fertility due to post-SMC complications, lack of partner consent, and other personal and religious beliefs [6,19].

Two studies from Uganda reported factors associated with SMC use among commercial motorcyclists including higher education, marital status, religion, alcohol consumption, perceived HIV risk, concern about being away from work, and a belief that SMC did not diminish sexual performance [6,19]. Commercial motorcyclists with a higher level of education, those who believed SMC did not reduce sexual pleasure, and those who were single had higher odds of using SMC services compared with those with primary education, those who believed that SMC diminished sexual performance, and who had separated from their sexual partner. Commercial motorcyclists who were concerned about being away from work because of SMC, those from the Christian faith, those who did not use alcohol, and those who perceived their risk for HIV was low had reduced odds of using SMC services compared with those who were not concerned about being away from work, from the Muslim faith, who used alcohol, and who perceived their risk of HIV infection was high [6,19].

## Interventions for HIV among commercial motorcyclists

Two studies from Nigeria reported on HIV interventions among motorcyclists (Table 3) [11,31]. A quasi-experimental behavior change intervention that involved training motorcyclists about HIV reported a change in HIV knowledge among commercial motorcyclists after 6 months, but no change in their attitudes or sexual practices [31]. A multifaceted intervention integrating structural (e.g., advocacy), behavioral (e.g., behavioral change communication), and biomedical strategies (e.g., provision of mobile VCT services, condom distribution) increased HIV services coverage (reach) to commercial motorcyclists in the study region [11].

Conversely, no HIV interventional studies were conducted among taxi drivers. Furthermore, no studies examined the effectiveness of HIV interventions in reducing HIV prevalence, improving antiretroviral therapy adherence, improving the use of other HIV prevention services, and mitigating HIV risk factors (e.g., substance use) among taxi drivers and

**Table 3. Intervention studies on HIV among commercial motorcyclists.**

| Author | Sample | Description of intervention | Outcomes measured | Findings |
|---|---|---|---|---|
| Olakunle et al. | Commercial motorcyclists | Behavior change intervention involving training motorcyclists on HIV | Knowledge of HIV symptoms, attitudes towards HIV, sexual behavioral practices | There was a statistically significant difference in knowledge scores (P<0.05) but not in attitudes or sexual practices. A statistically significant decrease in respondents with multiple sex partners was noted (P=0.01). |
| Adelekan et al. | Commercial motorcyclists | Multifaceted intervention integrating structural (advocacy), behavioral (behavioral change communication), and biomedical strategies (VCT, condoms, referral for antiretroviral therapy and sexually transmitted infections) | Coverage/ reach | The project was able to reach 76% (n=2878) and 73.5% (n=39,194) with VCT and condoms, respectively. |

commercial motorcyclists. For example, there was no interventional research on new models of HIV testing (e.g., HIV oral testing) and antiretroviral-based HIV prevention methods (e.g., PREP) [4].

## Discussion

This scoping review mapped the available evidence around HIV prevalence, risk factors for HIV infection, use of HIV prevention methods, and HIV interventions among taxi drivers and commercial motorcyclists in SSA. Most studies were conducted among men and commercial motorcyclists. The findings of this review revealed a high HIV prevalence in this group, ranging from 2.02% to 9.9% among commercial motorcyclists and up to 33.4% in samples including both commercial motorcyclists and taxi drivers. Our findings were comparable with the global burden of HIV among truck drivers, female sex workers (FSW), MSM, men in the general population, and men who paid for sex in SSA [2,32–34]. A systematic review [32] found the global burden of HIV among truck drivers was 3.86%, with the SSA region having the highest prevalence at 14.34%. UNAIDs reported the global prevalence of HIV among sex workers was 2.5%, which was within the range of our study findings [2]. In addition, other systematic review evidence indicated the pooled prevalence rates of HIV in SSA among MSM, men in the general population, and men who paid for sex were 17.81%, 6.15%, and 5.1%, respectively [33].

The HIV prevalence among taxi drivers and commercial motorcyclists in our study was much higher than that for adults globally (0.7%), adults in SSA (3%), and FSW and their clients in the Middle East and North Africa (1.4% and 0.4%, respectively) [2,35]. The difference between our study findings and the literature may be related to the fact that HIV prevalence is concentrated in certain groups (e.g., key populations such as MSM and FSW) and regions. Moreover, regions such as North Africa have traditionally had low HIV prevalence compared with the SSA region [35]. It is important to note that our findings regarding HIV prevalence are inconclusive because most results were based on self-reported data and might have been affected by recall, stigma, and social desirability biases. Importantly, no study investigated the HIV incidence among taxi drivers and commercial motorcyclists in SSA. Therefore, rigorous longitudinal research is necessary to investigate the HIV incidence and determine the HIV prevalence of this group in SSA based on robust laboratory methods and standard algorithms. This will provide comprehensive evidence on the burden of HIV in this key population in SSA.

The high HIV prevalence among taxi drivers and commercial motorcyclists in SSA may be linked to the numerous risk factors associated with HIV infections, as reported in our study. We found that the most HIV risk factors among taxi drivers and commercial motorcyclists were behavioral, followed by psychosocial and sociodemographic risk factors. Among behavioral risk factors, most taxi drivers and commercial motorcyclists had multiple sexual partners, were engaged in transactional sex, had condomless sex, and used substances such as alcohol during sexual intercourse. These behavioral HIV risk factors appear to be similar across all study participants irrespective of their sexual orientation and whether they belonged to a key population. For example, among long-distance truck drivers and fishing communities in Uganda,

multiple sexual partners, risky sexual networks, poor condom use, and illicit drug and alcohol use before or during sexual intercourse were reported to be the major drivers of HIV [32,36]. Despite condoms being an effective and low-cost HIV prevention method, our results revealed suboptimal condom use during sexual intercourse, which was consistent with condom use among men in SSA who paid for sex at their last sexual intercourse (68%) and FSW in the Middle East and North Africa (44%) [34,35]. The suboptimal condom use in SSA may be partly related to a reduced procurement, distribution, and social marketing of condoms by governments and donors in many low- and middle-income countries [2]. It is also important to note that despite similarities in behavioral risk factors between taxi drivers/commercial motorcyclists in SSA and other key populations, this HIV-priority population has received little attention and few interventions. There is a need for more efforts to reduce the HIV epidemic in this group.

Our review found that HIV knowledge, awareness, and perceived risk were high among taxi drivers and commercial motorcyclists. This concurred with a study among FSW in the Middle East and North Africa that found HIV knowledge and awareness was high (81.9%), although few FSW (34.6%) perceived themselves at high risk for HIV acquisition [35]. The difference between our findings and the literature may be related to variation in the study groups and tools used to quantify these concepts. Our results related to HIV knowledge, awareness, perceived risk, and stigma were inconclusive given the few studies that investigated these concepts and the use of unstandardized tools; HIV knowledge, awareness, perceived risk, and stigma may have been under- or over-estimated across studies. Importantly, few studies investigated sociodemographic risk factors, and no study explored the structural determinants of HIV infections among taxi drivers and commercial motorcyclists in SSA. Our findings concerning HIV risk factors may guide the design of effective HIV prevention and control interventions targeting taxi drivers and commercial motorcyclists in SSA. However, more rigorous cross-sectional and longitudinal research on HIV predictors is necessary (e.g., stigma and alcohol consumption). There is also a need to explore the structural determinants of HIV infections among taxi drivers and commercial motorcyclists in SSA as these factors directly influence the effective delivery of HIV prevention services. For example, there is need to explore how macrostructural-level factors, community-level factors, and work environmental factors influence HIV infection and prevention among taxi drivers and commercial motorcyclists in SSA.

Our findings revealed numerous gaps in the implementation of HIV services among taxi drivers and commercial motorcyclists, which are a priority and mobile HIV population. We found VCT was the most reported and used HIV prevention method (36.9%–96%) followed by SMC (20.7%–64.9%). This finding was higher than that reported in a study among FSW in the Middle East and North Africa, where only 17.6% of FSW had ever tested for HIV [35]. Therefore, addressing barriers to VCT and SMC access is paramount. Specifically, these findings indicated a need for innovative ways to provide multifaceted and targeted services to taxi drivers and commercial motorcyclists, as has been done among other mobile groups, such as FSW; for example, delivering HIV prevention methods (e.g., condoms, VCT services) near their workstations [2,37]. However, for such efforts to succeed, stakeholders need to engage and collaborate with transport workers' leadership. Furthermore, where services such as SMC and VCT are delivered to the general public, mechanisms to inform, support, and encourage commercial motorcyclists and taxi drivers to use those services via their leadership need to be in place.

There were numerous gaps in observational and interventional research related to HIV among commercial motorcyclists and taxi drivers in SSA. We found no interventional studies on the HIV cascade or effectiveness of HIV interventions in reducing HIV prevalence, improving antiretroviral therapy adherence, improving the use of other HIV prevention services, and mitigating HIV risk factors (e.g., substance use) among taxi drivers and commercial motorcyclists. In addition, there was no interventional research on new models of HIV testing (e.g., HIV oral testing) and antiretroviral-based HIV prevention methods (e.g., PREP) among taxi drivers and commercial motorcyclists. This lack of interventional research makes it difficult for stakeholders to understand effective approaches to mitigate the HIV epidemic among bridging populations. The lack of interventional research about certain HIV prevention methods (e.g., PREP) may be partly related to low integration and access to those services by high-risk groups (e.g.,

transport workers) in many low-income countries [2,4]. Furthermore, the lack of interventional research among taxi drivers and commercial motorcyclists in SSA (who mostly comprise men) may be related to the low attention given to men in the HIV cascade, where men are often treated as a homogenous population. This calls for a change in attitude among stakeholders and more rigorous research among motorcyclists and taxi drivers in SSA, a region with the highest HIV burden worldwide.

Finally, although the two interventional studies showed positive results such as increasing HIV knowledge, they were not rigorously designed, were conducted in one country (Nigeria), and were not scaled up. Such studies require scaling up in countries with high HIV rates such as Uganda, Kenya, and South Africa to enable researchers to measure their effectiveness in increasing HIV knowledge and improving attitudes, and practices towards HIV prevention in other settings across SSA. There is also a need to explore the role of peer-to-peer educators/counseling or group support models in reducing HIV infections, promoting the use of HIV prevention services (e.g., condom use, HIV testing services), and mitigating harmful HIV-related sexual behavior (through behavioral change communication) among this key population.

## Strengths and limitations

This is one of the first reviews to provide evidence for HIV infections among taxi drivers and commercial motorcyclists in SSA. Our findings may help stakeholders design further research and interventions to curb the HIV epidemic among bridging populations in SSA. This study also had limitations that should be considered. Most findings were inconclusive and most of the included studies were conducted among men and commercial motorcyclists. In other words, there was paucity of literature on taxi drivers and women operating in the transport systems of many SSA countries, indicating a major knowledge gap in HIV. The evidence described in this review best represents commercial motorcyclists. Therefore, because of geographical and participant bias in the included studies, our findings may not be generalizable to taxi drivers and women in many SSA countries. Most of the reviewed quantitative studies were cross-sectional and there were few cohort or interventional studies, meaning that our conclusions were based on non-rigorously conducted studies that conferred no causality. There is a need for further primary research with high rigor among taxi drivers and commercial motorcyclists in SSA. In addition, the lack of longitudinal predictors may make it difficult to inform the design of large-scale policies and programming. Therefore, conducting more studies in other countries using diverse populations of transport workers and rigorous designs is paramount. It is also difficult to make generalizations about the HIV risk factors mentioned in this review because of the various assessment strategies and tools used in the included studies. Additionally, due to a narrow focus, our study findings may not be generalizable to taxi drivers/ motorcyclists outside the SSA region. The durations of study publication also varied, meaning that research released in 2005 could be different from that published in 2024, potentially resulting in inconsistencies and ultimately bias within the findings and conclusions. Finally, although a rigorous search strategy was used, we could have missed some studies as we only included studies published in the English language.

## Conclusions

We found a high HIV prevalence among taxi drivers and commercial motorcyclists in SSA, which was above that in the global general adult population at 0.7%, but comparable with other key populations. This high HIV prevalence was associated with numerous behavioral, (e.g., multiple sex partners), psychosocial (e.g., stigma), and sociodemographic (e.g., age, income) risk factors. The use of HIV prevention methods such as SMC and condoms was suboptimal, and there were few interventional studies on HIV among the articles included in the review. Therefore, urgent action is needed in countries where taxi drivers and commercial motorcyclists play a significant role in the transport sector and may drive the HIV epidemic. Our findings indicate a need for innovative ways to provide multifaceted and targeted services to transport

workers as has been done for other mobile groups such as FSW. However, for the delivery of HIV services to succeed, stakeholders need to engage and collaborate with transport workers' leadership. Longitudinal research on HIV incidence and predictors, especially in exploring the structural determinants of HIV infections, is also needed as these factors have a direct influence on the effective delivery of HIV prevention services. There is also a need for interventional and observational research on new models of HIV testing (e.g., HIV oral testing) and antiretroviral-based HIV prevention methods (e.g., PREP) as well as the role of peer-to-peer educator or group support models in reducing HIV infections and mitigating harmful HIV related sexual behavior in this key population. These additional studies will provide more evidence that may provide comprehensive guidance for better HIV prevention and control policies and programming among taxi drivers and commercial motorcyclists in SSA.

## Supporting information

**S1 Table. Search strategy for article selection (PUBMED).**
(DOCX)

**S2 Table. Data extraction form-summary.**
(DOCX)

**S1 PRISMA-ScR Checklist. Preferred Reporting Items for Systematic reviews and Meta-Analyses extension for Scoping Reviews (PRISMA-ScR) Checklist.**
(DOCX)

## Acknowledgments

The authors express sincere gratitude to Ms. Audrey Holmes who copyedited a draft of this manuscript.

## Author contributions

**Conceptualization:** John Baptist Asiimwe, Benjamin Betunga.

**Formal analysis:** John Baptist Asiimwe, Benjamin Betunga, Lilian Birungi, Joy Kabasindi, Moses Wankiiri, Lilian Nuwabaine, Joseph Kawuki.

**Funding acquisition:** Joy Kabasindi.

**Investigation:** John Baptist Asiimwe.

**Methodology:** John Baptist Asiimwe, Benjamin Betunga, Lilian Birungi, Joy Kabasindi, Moses Wankiiri, Lilian Nuwabaine, Joseph Kawuki, Edward Kumakech.

**Project administration:** John Baptist Asiimwe, Benjamin Betunga, Lilian Birungi, Moses Wankiiri, Lilian Nuwabaine, Joseph Kawuki.

**Software:** John Baptist Asiimwe.

**Supervision:** John Baptist Asiimwe, Joseph Kawuki, Edward Kumakech.

**Validation:** John Baptist Asiimwe.

**Visualization:** Edward Kumakech.

**Writing – original draft:** John Baptist Asiimwe, Benjamin Betunga, Lilian Birungi, Joy Kabasindi, Moses Wankiiri, Lilian Nuwabaine, Joseph Kawuki, Edward Kumakech.

**Writing – review & editing:** John Baptist Asiimwe, Benjamin Betunga, Lilian Birungi, Joy Kabasindi, Moses Wankiiri, Lilian Nuwabaine, Joseph Kawuki, Edward Kumakech.

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
