## [Decision Letter · Decision Letter 0]

17 Apr 2025

PGPH-D-25-00089

HIV prevalence, risk factors for infection, use of prevention methods, and interventions among taxi drivers and commercial motorcyclists in sub-Saharan Africa: A scoping review

Dear Dr. Asiimwe,

Thank you for submitting your manuscript to PLOS Global Public Health. After careful consideration, we feel that it has merit but does not fully meet PLOS Global Public Health’s publication criteria as it currently stands. Therefore, we invite you to submit a revised version of the manuscript that addresses the points raised during the review process.

We look forward to receiving your revised manuscript.

Kind regards,

Kavi Bhalla

Academic Editor

Journal Requirements:

1. Please upload a copy of Figures 1,2,3 which you refer to in your text on page 8, 11, 12. Or, if the figure is no longer to be included as part of the submission please remove all reference to it within the text. 2. Please provide separate figure files in .tif or .eps format. For more information about figure files please see our guidelines:  LINKhttps://journals.plos.org/globalpublichealth/s/figures https://journals.plos.org/globalpublichealth/s/figures#loc-file-requirements

Additional Editor Comments (if provided):

Please make the modifications suggested by the reviewers.

Reviewers' comments:

Reviewer's Responses to Questions

**Comments to the Author**

1. Does this manuscript meet PLOS Global Public Health’s publication criteria ? Is the manuscript technically sound, and do the data support the conclusions? The manuscript must describe methodologically and ethically rigorous research with conclusions that are appropriately drawn based on the data presented.

Reviewer #1: Yes

Reviewer #2: Yes

Reviewer #3: Yes

2. Has the statistical analysis been performed appropriately and rigorously?

Reviewer #1: N/A

Reviewer #2: Yes

Reviewer #3: Yes

3. Have the authors made all data underlying the findings in their manuscript fully available (please refer to the Data Availability Statement at the start of the manuscript PDF file)?

Reviewer #1: Yes

Reviewer #2: Yes

Reviewer #3: Yes

4. Is the manuscript presented in an intelligible fashion and written in standard English?

Reviewer #1: Yes

Reviewer #2: Yes

Reviewer #3: Yes

5. Review Comments to the Author

Reviewer #1: The reviewed research article demonstrates a rigorous methodological process for the design related to a scoping review. As an exploratory study, it allows for the identification of key aspects concerning HIV prevalence, risk factors for HIV infection, the use of HIV prevention methods, and HIV interventions among taxi drivers and commercial motorcyclists in sub-Saharan Africa. It explicitly outlines the limitations present in the study.

The following adjustments must be made:

Line 41 y 129:

I suggest indicating the start date of the search (2000).

Line 179:

Study flow: adjust the text corresponding to the articles that have been classified as qualitative research, as of the five articles referenced, the only qualitative one is “Mchunu GG, Naidoo JR, Ncama BP. Condom use: a less travelled route among minibus taxi drivers and their taxi queens in KwaZulu-Natal, South Africa. Afr Health Sci. 2020;20(2):658-65.”

Line 195:

Figure 1. Study selection process. Adjust the information related to the “studies included in qualitative synthesis” and “studies included in quantitative synthesis” boxes, as not all the studies have a meta-analysis design.

Line 209:

Table 2. Characteristics of the studies included in this scoping review. Adjust the Study design column for the articles classified as qualitative research, as some are Mixed or Quantitative.

Line 251:

Figure 3. Conceptual framework. Cite the theoretical sources supporting the proposal.

Line 605:

The phrase “among the study group” - I recommend clarifying that it refers to the articles included in the review.

Line 617:

Based on the results presented in this document, review the following conclusion and its scope: “These studies will provide more evidence that may provide comprehensive guidance for better HIV prevention and control policies and programming among taxi drivers and commercial motorcyclists in SSA.”

Reviewer #2: Thank you for the opportunity to review the manuscript titled ‘HIV prevalence, risk factors for infection, use of prevention methods, and interventions among taxi drivers and commercial motorcyclists in sub-Saharan Africa: A scoping review.’

Overall, the manuscript is fairly well written. However, I suggest the following comments and clarifications to improve its quality.

1. Title: The title is too long. I would suggest shortening it to capture the key theme that defines the entire study.

2. Abstract:

3. Background: Quote some real figures on HIV prevalence among taxi drivers to show the magnigute of the problem in this population.

4. Methods: Line 41; Change inception to the actual year. The word inception may leave one wondering from when exactly?

5. Results: State the total number of articles initially screened before stating the total number that was finally included.

6. Conclusion: The conclusion seems to deviate from the main objective of the study; state what was the HIV prevalence, the risk factors and prevention methods identified from the current study. What is given is more of implications and recommendations rather than an actual conclusion based on the study objectives and results.

7. Introduction:

8. I would suggest introducing the global perspective of HIV before Sub-Saharan Africa in the first paragraph line 70.

9. I would suggest quoting some figures on actual HIV prevalence rates reported in other studies in the first paragraph, to highlight the magnitude of the condition among taxi drivers.

10. A justification on why the study focused only on Sub-Saharan Africa is needed. Taxi drivers in others parts of the world are equally at risk of HIV. A global perspective would have made the evidence produced in this study more stronger.

11. Methodology:

12. Line 129: From inception to 2024; State a specific year. ‘Inception’ may not be very clear on which year you are reffering to.

13. Line 167: Cite the SPSS software used in the analysis.

14. Line 168: Elaborate on the qualitative data highlighted here. How did you get these data from the included studies? Did you pick the quotes reported in the studies? The approach used in getting the qualitative data is not very clear.

15. Results:

16. Highlight how many studies were identified from the initial search of each individual database.

17. Discussion

18. Discuss to acknowledge how the variation in the periods of study publication may have affected the overall quality of your study. Also the Limitations section should take into consideration how studies published in 2005 may differ from those published in 2024, and how this affects the overall evidence reported here.

19. Conclusuion:

20. Quote the actual ‘high’ prevalence that you identified from your study. High is relative not very specific. Also, quote the HIV prevalence in the general adult population for a more clear comparison with that of the taxi-drivers.

21. Line 603-604; highlight a few of the behavioral, psychosocial, and sociodemographic risk factors identified in this study.

Reviewer #3: The manuscript titled "HIV prevalence, risk factors for infection, use of prevention methods, and interventions among taxi drivers and commercial motorcyclists in sub-Saharan Africa: A scoping review" addresses a significant public health concern by exploring the various aspects of HIV in these vulnerable populations. The research presented is robust and highly relevant, highlighting crucial issues that can help inform policies and interventions aimed at reversing HIV trends in sub-Saharan Africa. However, there are areas that need clarification, including terminology consistency, the categorization of studies, and alignment between statements and data presented. Overall, the authors have made a commendable contribution to the field, and with some revisions, the manuscript can provide even more impactful insights.

Kindly see attached file with my comments

6. PLOS authors have the option to publish the peer review history of their article (what does this mean? ). If published, this will include your full peer review and any attached files.

**Do you want your identity to be public for this peer review?** For information about this choice, including consent withdrawal, please see our Privacy Policy .

Reviewer #1: No

Reviewer #2: **Yes: ** CYRUS MUTIE

Reviewer #3: **Yes: ** Berrick Otieno

---

## [Editor Report · Decision Letter 1]

1 May 2025

HIV prevalence, risk factors, prevention methods, and interventions among taxi drivers and commercial motorcyclists in sub-Saharan Africa: A scoping review

PGPH-D-25-00089R1

Dear Mr., Asiimwe,

We are pleased to inform you that your manuscript 'HIV prevalence, risk factors, prevention methods, and interventions among taxi drivers and commercial motorcyclists in sub-Saharan Africa: A scoping review' has been provisionally accepted for publication in PLOS Global Public Health.

Best regards,

Kavi Bhalla

Academic Editor